# A Predictive Model for Real–time Prediction of Intradialytic Hypotension Based on Machine Learning Algorithms

### Yuping Jiang
Shentaiwang Health Technology (Nanjing) Co., Ltd.
Nanjing, China
ypjiang@stwitinc.com

### Xueqin Bian
The Second Affiliated Hospital of Nanjing Medical University
Nanjing, China
bianxueqin@njmu.edu.cn

### Xueming Hong
Shentaiwang Health Technology (Nanjing) Co., Ltd.
Nanjing, China
xmhong@stwitinc.com

### Haiyan Si
Shentaiwang Health Technology (Nanjing) Co., Ltd.
Nanjing, China
hysi@stwitinc.com

### Di Liu
Shentaiwang Health Technology (Nanjing) Co., Ltd.
Nanjing, China
dliu@stwitinc.com

### Hao Chen[1]
Shentaiwang Health Technology (Nanjing) Co., Ltd.
Nanjing, China
hchen@stwitinc.com

## ABSTRACT

**Objective** Develop a machine learning-based model to predict IDH using pre-dialysis features. And to continuously predict IDH within the next hour during the dialysis session by incorporating real-time monitoring data. This approach helps in timely intervention, potentially reducing IDH rates and improving clinical outcomes for patients.

**Methods** Collected maintenance hemodialysis (MHD) patients from October 1, 2021, to July 31, 2022, and divide them into development and validation datasets based on the treatment time point of May 1, 2022. IDH is defined as follows: (1) Nadir90: intradialytic systolic blood pressure (SBP) < 90 mmHg; (2) Fall20Nadir90: intradialytic SBP < 90 mmHg and a drop of ≥ 20 mmHg from pre-dialysis SBP. Analyzed the model's predictive performance trained with various machine learning (ML) classification algorithms using k-fold cross-validation, evaluated by plotting the receiver operating characteristic curve (ROC) and precision-recall curve (PRC), calculating the area under the ROC (AUROC) and PRC (AUPRC), and computing the true positive rate (TPR), and true negative rate (TNR).The XGBoost algorithm was used to identify the important features required for the warning models.

**Results** Data from 644 patients were analyzed, contributing 61,823 HD sessions with 302,942 intradialytic SBP measurements. IDH occurred in 2,659 (4.3%) HD sessions (Nadir90), in 1,706 (2.76%) sessions (Fall20Nadir90). Among various models compared, XGBoost achieved the best performance for predicting IDH before HD session (TPR: 0.6, TNR: 0.99, AUROC: 0.955, AUPRC: 0.686). Key predictive features included historical minimum SBP, average of historical minimum SBP, current SBP, diastolic blood pressure (DBP), IDH incidence rate, interdialytic weight change rate, prescribed dialysis duration, and dialysis vintage. The real-time model for predicting IDH within the next hour showed a TPR of 0.89, TNR of 0.92, AUROC of 0.959, and AUPRC of 0.38, with additional important features being mean arterial pressure (MAP), dialysis time, and ultrafiltration (UF) changes.

**Conclusion** The XGBoost model has a high predictive capability for IDH during an ongoing HD session, assisting healthcare providers in assessing IDH risk and making timely decisions.[1]

## KEYWORDS

Hemodialysis, intradialytic hypotension, machine learning, predicting model, XGBoost

## 1 INTRODUCTION

Intradyalytic hypotension (IDH) is a common and serious complication during hemodialysis (HD), with significant risk implications. IDH not only affects the quality of life of patients but can also lead to severe cardiovascular complications and even death[1][2]. According to a study on the mechanisms of IDH[3], the prevalence of IDH during hemodialysis is approximately 10-12%. Additionally, a recent comparative study[4] on the correlation between IDH and increased mortality

*WOODSTOCK'18, June, 2018, El Paso, Texas USA*
© 2018 Copyright held by the owner/author(s). 978-1-4503-0000-0/18/06...$15.00
https://doi.org/10.1145/1234567890

[1] Corresponding authors' E-mail address: hchen@stwitinc.com;
Tel: 86-025-85567902

evaluated different definitions of IDH and found its incidence 11.19% - 21.7% of the sessions, and 32.39% - 56.34% of the patients. Traditionally, methods to prevent and treat IDH have been varied[5][6], such as reducing the ultrafiltration (UF) rate, avoiding significant interdialytic weight gain, increasing weekly treatment time, adjusting the soac output, impaired vascular resistance, and physiological parameters during HD. However, the rapid advancement of artificial intelligence (AI) and machine learning (ML) in recent years has intdium concentration and temperature of the dialysate, and avoiding food intake during dialysis. These methods mainly rely on manual intervention by healthcare professionals. This approach is not only time-consuming and labor-intensive but may also lead to suboptimal treatment outcomes due to untimely or inappropriate interventions[7]. Therefore, developing accurate and real-time IDH prediction models is of great significance for improving the prognosis of HD patients.

The influencing factors of IDH include blood pressure, weight, hemoglobin level, blood glucose level, excessive UF, reduced cardiroduced new prospects for IDH research[8][9], and achieving more accurate and real-time IDH prediction[10][11][12].

Lee[7] et al. developed a deep learning model using pre-dialysis clinical variables to predict IDH. Allinovi[13] et al. emphasized the assessment of patients' fluid status before dialysis, using non-invasive imaging techniques to provide crucial predictive information. Zhang[14] et al. focused on utilizing real-time intra-dialysis data, employing ML algorithms to analyze real-time blood pressure and other dynamic variables during the dialysis process, achieving high predictive accuracy. Li[15] et al. used the dynamic changes in intra-dialysis data, employing advanced optimization algorithms and ML models for real-time prediction. Kim[16] et al. utilized deep learning techniques to perform high-precision analysis on real-time data, with particular attention to processing time-series data. Our experiments have demonstrated that both pre-dialysis information and intra-dialysis data hold significant value in predicting IDH.

In this study, we developed an early warning model using pre-dialysis information and incorporated intra-dialysis data to enhance prediction accuracy. And we simplified the model while maintaining predictive accuracy. This approach not only improved the model performance but also ensured real-time prediction capability.

## 2 MATERIALS AND METHODS

### 2.1 Study Population

We conducted a retrospective study on HD patients at the Second Affiliated Hospital of Nanjing Medical University from October 1, 2021, to July 31, 2022. The data from 71,664 HD sessions if 1,274 HD patients. The exclusion criteria for HD patients were: (1) Age < 18 years; (2) Insufficient basic information and clinical history; (3) HD treatment duration less than three months. Exclusion criteria for HD sessions were: (1) No laboratory test data available in the past 3 months; (2) Blood pressure monitoring intervals during dialysis exceeding 1.5 hours; (3) Pre-dialysis blood pressure not recorded (Figure 1).

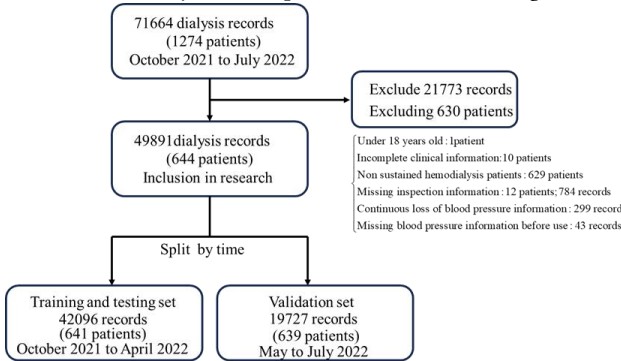

Figure 1. Data Screening and Dataset Partitioning. Screening involves filtering patients and dialysis records, and the dataset is partitioned based on time.

In the end, we included data from 49,891 HD sessions of 644 patients. Since this project was conducted as an internal quality improvement initiative for the HD system, it did not undergo institutional review board (IRB) review. We divided the dataset based on the dialysis treatment timestamp on May 1, 2022. The data before this date were used for training and testing the model, while the remaining data were used for validating the model performance. The training and testing sets were randomly split in an 8:2 ratio, and a 5-fold cross-validation was applied.

### 2.2 Hemodialysis Sessions

The monitored data of each HD session is automatically saved to the HD system database. Apart from the initial HD script setup, monitored data are collected every 20 seconds from the HD machines, including arterial line pressure (AP), venous line pressure (VP), blood flow rate, dialysate flow rate, UF rate, total UF volume, dialysate temperature, and conductivity. Vital signs, including SBP, DBP, MAP, and pulse rate, are recorded by default every hour and aligned with the time points of SBP measurements from the HD machine. Additional blood pressure measurements are taken when patients complain of any symptoms related to blood pressure abnormalities.

### 2.3 Study Outcomes

We defined IDH events: (1) Nadir90: intra-dialysis SBP < 90 mmHg; (2) Fall20Nadir90: intra-dialysis SBP < 90 mmHg, a decrease in SBP ≥ 20 mmHg compared to pre-dialysis SBP. Each IDH definition was treated as a separate binary outcome.

### 2.4 Study Variables and Data Processing

The dataset for this study comprises any data recorded by the HD system during the HD treatment process (Supplementary data, Table S1). It includes clinical baseline information, medical history, historical HD sessions, current pre-dialysis examinations,

laboratory test results (of each HD treatment in the past three months.), and pre-dialysis vital signs, a total of 120 features.

Additionally, we constructed new features from the data, including changes in monitoring data during the last HD session, statistics of IDH events and changes in monitoring data over the past week and the past month, changes in monitoring data during the current HD session, and the relationship between UF volume and body weight and dry weight. Specifically, The additional features and their construction methods are shown in Supplementary (Supplementary data, table S2)

## 2.5   Model Development and Validation

The SBP monitoring frequency during HD sessions is once per hour, allowing us to calculate the risk of IDH in the next hour in real-time.

Using both training and testing datasets, we trained and tested models using Multilayer Perceptron (MLP)[17], K-Nearest Neighbors (KNN), Support Vector Machine (SVM), Random Forest (RF), and XGBoost algorithms. To ensure the robustness of model performance, we employed 5-fold cross-validation to analyze and compare the predictive performance of models on the testing dataset. We selected the algorithms with better performance and comprehensively evaluated and analyzed the predictive performance of the models on the validation dataset.

The model evaluation metrics include: positive predictive value, negative predictive value, true positive rate (TPR), true negative rate (TNR), F1 score, AUPRC, and AUROC. Additionally, we plotted ROC and PRC (Table 1).

### Table 1: Average Metrics of Predictive Models

| Evaluation Metrics | Computational Methods | Elucidation |
|---|---|---|
| Positive Predict Value | $PPV = TP/(TP+FP)$ | The probability that an individual who tests positive is truly at risk for IDH. |
| Negative Predict Value | $NPV = TN/(FN+TN)$ | The probability that an individual who tests negative is truly not at risk for IDH. |
| True Positive Rate | $Se = TP/(TP+FN)$ | The probability that an individual who is actually at risk is diagnosed as at risk by the test. |
| True Negative Rate | $Sp = TN/(TN+FP)$ | The probability that an individual who is actually not at risk is diagnosed as not at risk by the test. |
| Youden Index | $YI = (Se+Sp)-1$ | The accuracy index. |
| F1 | $2*(PPV*Se)/(PPV+Se)$ | The harmonic mean of the positive predictive value and the true positive rate. |

## 2.6   Model Interpretation

We employ Shapley Additive exPlanation (SHAP)[18] to assess the impact of the model on predicting IDH. This is done to gain a deeper understanding of each variable's contribution to the prediction results. SHAP, as a highly interpretable tool, provides an accurate and transparent method to clearly illustrate the effect of each feature on the predictions across different samples. Through SHAP, we can gain a deeper insight into the logic behind the model's decisions, thereby more accurately predicting the risk of IDH.

## 3   RESULTS

## 3.1   Characteristics of Hemodialysis Sessions

We conducted our study analysis on data from 49,891 HD sessions of 644 patients, with a total of 302,942 intra-dialysis SBP. The demographic statistics are summarized in Table 2, with patients having a mean age of 58 ± 13 years and a mean dialysis vintage of 93.4 ± 84.6 months. The majority of patients were male (62.9%), and most had comorbidities such as hypertension (72.5%) and diabetes (31.4%). The primary causes of chronic kidney disease (CKD) were hypertensive nephropathy (32.6%), glomerulonephritis (28.7%), and diabetes nephropathy (25.9%). Among the 644 patients, 43.5% experienced at least one episode of IDH, with at least 4.3% of HD sessions resulting in an IDH event. (Supplementary data, table S3)

### Table 2 Basic Information Statistics of Included Patients

| Statistical Project | Statistics Results | Statistical Project | Statistics Results |
|---|---|---|---|
| Age, years [mean (SD)] | 58.0(13.0) | Medical History [n (%)] | |
| Dialysis age, months [mean (SD)] | 93.4(84.6) | Hypertension | 467(72.5) |
| Male [n (%)] | 405(62.9) | Diabetes | 202(31.4) |
| Primary Disease [n (%)] | | Parathyroidectomy | 117(18.2) |
| Hypertensive Kidney Damage | 210(32.6) | Cardiovascular Disease | 61(9.5) |
| Glomerulonephritis | 185(28.7) | Cerebrovascular Disease | 38(5.9) |
| Diabetes Nephropathy | 167(25.9) | Cerebral Infarction | 19(3) |
| Autosomal Dominant Polycystic Kidney Disease | 25(3.9) | Type of Vascular Access [n (%)] | |
| Other | 29(4.2) | TCC | 76(11.8) |
| Unknown Primary Disease | 66(10.2) | AVF | 491(76.2) |
| Infection [n (%)] | | AVG | 62(9.6) |
| Hepatitis C | 120(18.6) | NTC | 15(2.4) |
| Hepatitis B | 78(12.1) | | |
| Syphilis | 36(5.6) | | |

TCC: Tunneled Cuffed Central Venous Catheter; AVF: Autogenous Arteriovenous Fistula; AVG: Arteriovenous Graft; NTC: Non-tunneled Non-cuffed Catheter;

We divided 61,823 HD sessions based on the treatment time point of May 1, 2022. The training and testing dataset: 42,096 HD sessions from 641 patients, the validation dataset: 19,727 dialysis records from 639 patients.

## 3.2   Model Performance

### Pre-dialysis IDH prediction

The XGBoost model outperformed other algorithm models in terms of AUROC (Nadir90: 0.939, Fall20Nadir90: 0.924) (Table 3 and Supplementary data, figure S1). Finally, this model was selected for statistical analysis of its predictive performance on the validation set.

Table 3 the comparative results of 5-fold cross-validation for multiple algorithm models.

| model | Nadir90 | | Fall20Nadir90 | |
|---|---|---|---|---|
| | AUROC | AUPRC | AUROC | AUPRC |
| MLP | 0.733 (0.662-0.847) | 0.554 (0.396-0.741) | 0.612 (0.522-0.686) | 0.442 (0.34-0.548) |
| KNN | 0.686 (0.569-0.781) | 0.511 (0.287-0.694) | 0.600 (0.552-0.647) | 0.361 (0.258-0.454) |
| SVM | 0.631 (0.520-0.743) | 0.528 (0.334-0.717) | 0.500 (0.500-0.500) | 0.514 (0.506-0.520) |
| RF | 0.907 (0.867-0.943) | 0.573 (0.356-0.787) | 0.906 (0.858-0.934) | 0.385 (0.205-0.532) |
| XGBoost | 0.939 (0.905-0.967) | 0.563 (0.313-0.799) | 0.924 (0.885-0.946) | 0.392 (0.173-0.523) |

The model performance on the validation set: (1) Nadir90: an AUROC of 0.955 [95% CI 0.947-0.962] (Figure 2A), with an AUPRC of 0.686 (Supplementary data, figure S2A); (2) Fall20Nadir90: an AUROC of 0.933 [95% CI 0.922-0.945] (Figure 2B), with an AUPRC of 0.440 (Supplementary data, figure S2B).

For Nadir90, Following discussions with healthcare professionals, a 10% false positive rate is considered acceptable[7]. Therefore, we selected the threshold of 0.05, corresponding to a TNR of 0.9, as the classifying threshold for IDH probabilities. This threshold corresponds to the maximum Youden's index, and both TPR and TNR reach 0.90 (Supplementary data, table S4). The threshold corresponding to the maximum F1 score for classifying IDH probabilities is 0.48, with TPR and TNR of 0.60 and 0.99, respectively.(Supplementary data, figure S3A)

Similarly, for the Fal20lNadir90, the performance analysis of the prediction model at each threshold is shown in Table S5 and Supplementary data, figure S3B.

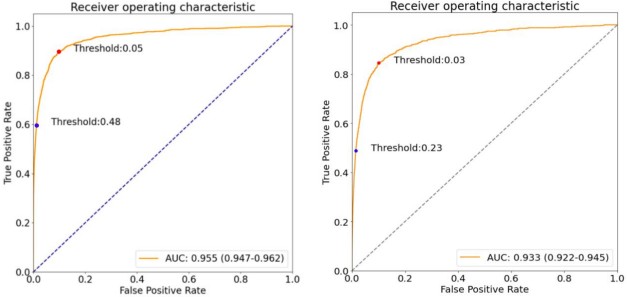

Figure 2. The red and blue dots on the ROC and PRC curves respectively represent the threshold points corresponding to the maximum Youden's index and maximum F1 value.

**Intra-dialysis IDH prediction**

For Nadir90, an AUROC of 0.963 [95% CI 0.956-0.970] (Supplementary data, figure S4A), with an AUPRC of 0.389 (Supplementary data, figure S4B). We selected the threshold of 0.02, corresponding to a TNR of 0.89 and TPR of 0.93, as the classifying threshold for IDH probabilities. While, we selected the threshold of 0.27, corresponding to a TNR of 0.99, as the classifying threshold for IDH probabilities, meeting the requirement that the FPR is less than 10%. This threshold

corresponds to the maximum F1-score. (Supplementary data, table S6 and Supplementary data, figure S5)

## 3.3 Variable Importance

Supplementary data, figure 6 illustrates the importance of each variable in the pre-dialysis IDH prediction model, plotted as a bar graph based on the average SHAP values. The results indicate that the monthly minimum SBP, monthly average low SBP, weekly average SBP, and weekly minimum SBP have the greatest impact on the model predictions. Monitored SBP, DBP, the incidence rate of IDH events, dialysis duration, and dialysis vintage also significantly affect the prediction results, suggesting their relevance to IDH.

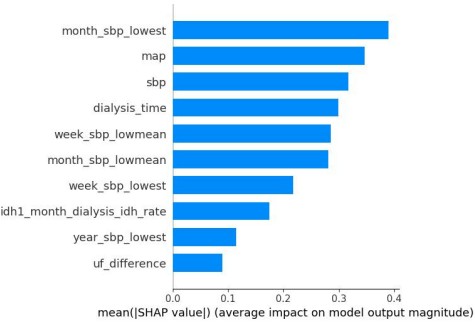

Figure 3. The XGBoost model predicts the top 10 important features of the intra-dialysis IDH (Nadir90)

In addition, the important characteristics of the intra-dialysis IDH prediction model are : Monitored monthly minimum SBP, map, SBP, weekly average low SBP, monthly average low SBP, weekly low SBP, yearly low SBP, difference between the UF, dialysis time, the incidence rate of IDH events.

## 4 DISCUSSION

The definition of IDH typically involves the nadir SBP and its decrease from pre-dialysis SBP. Considering the association between different definitions of IDH and mortality risk[2], this study defines IDH as Nadir90 and Fall20Nadir90.

In this study, we utilized both pre-dialysis and intra-dialysis data to train a lightweight predictive model, achieving accurate prediction of IDH. By comprehensively leveraging pre-dialysis clinical variables and real-time dynamic data during dialysis, we significantly improved the model's predictive performance using XGBoost algorithm. Our model was developed and validated on a diverse patient cohort comprising 61,823 HD sessions. The pre-dialysis model's performance for Nadir90 achieved an AUROC of 0.955 and an AUPRC of 0.686, while for Fall20Nadir90, it achieved an AUROC of 0.933 and an AUPRC of 0.440. The intra-dialysis warning for Nadir90 had an AUROC of 0.959 and an AUPRC of 0.38. The FPR corresponding to the maximum Youden's index and maximum F1 score were both less than 10%. Compared to models using only a single type of data, our approach can more comprehensively capture patients' health

status and physiological changes, thereby improving prediction accuracy and reliability. Additionally, our model predicts two different definitions of IDH events, providing crucial support for personalized prediction and intervention.

We divided the sample data based on specified dialysis treatment time points. This approach simulates the IDH warning system's real-world performance. In contrast, a validation set created by randomly splitting the sample data would share the same distribution as the development and testing sets, making the validation results less representative of real-world usage.

ML methods, by leveraging various types of data, can more comprehensively capture these complex factors, thereby achieving more accurate predictions. We compared the models performance of several algorithmic models, including MLP, KNN, SVM, RF, and XGBoost. Among these, the model trained with the XGBoost algorithm exhibited the best performance, and supported the calculation of risk probabilities even in the presence of missing feature values. Currently, many studies use ML methods to predict IDH, but most rely solely on either pre-dialysis or intra-dialysis data[7][10]. Our research distinguishes themself by integrating pre-dialysis and intra-dialysis data, achieving superior predictive performance.

Pre-dialysis data (such as the patient's baseline health status and initial physiological parameters) provide an assessment of the patient's initial health condition and potential risks. Intra-dialysis data (such as real-time blood pressure and other dynamic physiological variables during the dialysis process) reflect the patient's dynamic physiological changes during dialysis. Considering the need for real-time predictions, we have streamlined the model. This improvement ensures not only high prediction accuracy but also the model's real-time capabilities.

One notable limitation of our data is the lack of records on oral medications taken during dialysis treatment, which prevents us from fully accounting for individual medication regimens when predicting the risk of IDH. Additionally, since our data are from a single center, we cannot perform multi-center validation, thereby limiting our ability to verify the generalizability of the IDH prediction model.Finally, our model shares a common drawback with many ML models: it is challenging to interpret how the model arrives at its predictions or how individual factors influence the outcomes[19].

In summary, ML algorithm-based models can predict the risk of IDH occurrence both before and during dialysis. Further prospective studies are needed to evaluate whether this predictive information helps healthcare professionals intervene early and prevent IDH events, as well as to assess the improvement in IDH event rates and patient outcomes.

## ACKNOWLEDGMENTS

The authors express gratitude to the nephrologist at the Second Affiliated Hospital of Nanjing Medical University for their valuable advice.

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

## Supplementary data

Table S1: Data Collection Categories and Specific Features

| Category | Feature |
|---|---|
| Demographic Characteristics | Age、Gender、Ethnicity; |
| Basic Information | Height、Blood Type、Duration of Dialysis、Vascular Access and Type、Age at First Dialysis; |
| Medical History Information | Primary Disease、Infection History、Significant Medical History; |
| Laboratory and Examination Information | Hemoglobin、Mean Corpuscular Hemoglobin Concentration (MCHC)、Mean Corpuscular Hemoglobin (MCH)、Platelet Count、White Blood Cell Count、Red Blood Cell Count、Hematocrit、Mean Corpuscular Volume (MCV)、Creatinine、Urea、Uric Acid、Albumin、Total Protein、Prealbumin、Glucose、Sodium、Potassium、Calcium、Phosphorus、Chloride、Bicarbonate、Plasma Fibrinogen、Fibrin(ogen) Degradation Products (FDP)、Plasma D-Dimer、Serum Cardiac Troponin I、Serum Cardiac Troponin T、Myoglobin、Serum Iron、Total Iron-Binding Capacity (TIBC)、Serum Ferritin、Parathyroid Hormone (PTH)、Urine Occult Blood、Urine Protein、Normalized Protein Catabolic Rate (nPCR)、Kt/V、URR; |
| Medical Record Information | Comorbidities (Outpatient Diagnosis、Discharge Diagnosis) (mainly including Anemia、Cancer、Arrhythmia、Cerebrovascular Disease、Chronic Obstructive Pulmonary Disease、Diabetes、Gastrointestinal Bleedin、Hyperparathyroidism、Infection During Treatment、Cardiovascular Disease、Peripheral Vascular Disease、Hypertension); |
| Historical Dialysis Plans | Dialyzer Type、Dialysis Mode、Replacement Method、Dialysis Time、Ultrafiltration Volume、Anticoagulant Category、Anticoagulant Dosage、EPO Dosage、Calcium Supplement Dosage、Dialysate Temperature、Dialysate Electrolyte Concentration (Ca、K、Na)、Conductivity、Blood Flow Rate、Last Dialysis Machine Weight; |
| Historical Pre-dialysis Physical Examination Information | Dry Weight、Pre-dialysis Weight、Pre-dialysis Systolic Blood Pressure、Pre-dialysis Diastolic Blood Pressure、Mean Arterial Pressure、Pre-dialysis Heart Rate、Pre-dialysis Body Temperature; |
| Historical Intra-dialysis Monitoring Information | Intradialytic Monitoring ID、Intradialytic Monitoring Time、Intradialytic Systolic Blood Pressure、Intradialytic Diastolic Blood Pressure、Mean Arterial Pressure、Intradialytic Heart Rate、Intradialytic Body Temperature、Ultrafiltration Achieved、Venous Pressure、Transmembrane Pressure、Blood Flow Rate; |
| Historical Post-dialysis Summary | Post-dialysis Systolic Blood Pressure、Post-dialysis Diastolic Blood Pressure、Mean Arterial Pressure、Post-dialysis Heart Rate、Post-dialysis Body Temperature、Post-dialysis Ultrafiltration、Coagulation Status、Hematoma Situation、Occurrence of IDH (Intradialytic Hypotension)、Other Symptoms During Dialysis; |
| Current Dialysis Plan | Dialysis Record ID、Dialyzer Type、Dialysis Mode、Replacement Method、Dialysis Time、Ultrafiltration Volume、Anticoagulant Category、Anticoagulant Dosage、EPO Dosage、Zocor、Calcium Supplement Dosage、Other Medications、Dialysate Temperature、Dialysate Electrolyte Concentration (Ca、K、Na)、Conductivity、Blood Flow Rate、Previous Post-dialysis Weight; |
| Current Pre-dialysis Physical Examination Information | Dry Weight、Pre-dialysis Weight、Pre-dialysis Systolic Blood Pressure、Pre-dialysis Diastolic Blood Pressure、Mean Arterial Pressure、Pre-dialysis Heart Rate、Pre-dialysis Body Temperature、BMI; |
| Current Intra-dialysis Monitoring Information | Intradialytic Monitoring ID、Intradialytic Monitoring Time、Intradialytic Systolic Blood Pressure、Intradialytic Diastolic Blood Pressure、Mean Arterial Pressure、Intradialytic Heart Rate、Intradialytic Body Temperature、Ultrafiltration Achieved、Venous Pressure、Transmembrane Pressure、Blood Flow Rate、Dialysate Temperature; |

Note:nPCR = (Pre dialysis Blood Urea Nitrogen - Post dialysis Blood Urea Nitrogen) × (0.045/2 Days between two blood samples)

Table S2 The additional features and their construction methods

| No. | Features construction methods |
|---|---|
| 1 | For the monitoring data from the last dialysis session, we calculated the minimum and average values of systolic blood pressure (SBP), diastolic blood pressure (DBP), and heart rate. Additionally, for the dialysis data from the past week and the past month, we calculated the number of dialysis sessions, dialysis frequency, number of IDH events, frequency of IDH events, as well as the minimum SBP during dialysis, and the minimum and average values of SBP, DBP, mean arterial pressure (MAP), and heart rate. |
| 2 | During intra-dialytic predictive modeling, new features were constructed using monitoring data. These included systolic blood pressure (SBP), diastolic blood pressure (DBP), mean arterial pressure (MAP), heart rate, UF volume, monitoring time (used to calculate dialysis duration), dialysate temperature, and blood flow rate. For these indicators, we calculated the difference from pre-dialysis values, the difference between consecutive monitoring data points, the rate of change, and the difference in the rate of change, to represent the changes in these indicators. |
| 3 | Additionally, we calculated the weight change during the interdialytic period ((current pre-dialysis weight - dry weight) and (current pre-dialysis weight - last post-dialysis weight)), interdialytic weight change rate ((current pre-dialysis weight - last post-dialysis weight) / dry weight), current prescribed dehydration volume / dry weight, UF target rate (UF target rate), and UF rate as new features. The UF rate is calculated as UF volume (ml) / dialysis time (h) / dry weight (kg). |

The formulas for calculating the UF target rate and DF target is as follows:

$$\text{UF target rate} = \frac{(pre-dialysis\ weight(kg)) - (dry\ weight(kg))}{(treatment\ time(hour)) \times (dry\ weight(kg))}$$

$$\text{UF rate} = \frac{(ultrafiltration\ volume(L))}{(treatment\ time(hour)) \times (dry\ weight(kg))}$$

Table S3 Dialysis Records and IDH Statistics of Included Patients

| Statistical Project | Overall | Training and Testing Set | Validation Set |
|---|---|---|---|
| Number of patients | 644 | 641 | 639 |
| Number of people who have experienced IDH-Nadir90 (n,%) | 280(43.48) | 244(38.07) | 156(24.41) |
| Number of people who have experienced IDH-Fall20Nadir90 (n,%) | 256(39.75) | 224(34.95) | 139(21.75) |
| Number of dialysis treatments | 61,823 | 42,096 | 19,727 |
| Number of dialysis treatments with IDH_1adir90 occurrence (n,%) | 2,659(4.3) | 1,859(4.42) | 800(4.06) |
| Number of dialysis treatments with IDH-Fall20Nadir90 occurrence (n,%) | 1,706(2.76) | 1,242(2.95) | 464(2.35) |
| Number of times blood pressure is monitored during dialysis | 302,942 | 206,273 | 96669 |

Table S4 the performance analysis of the pre-dialysis IDH prediction model at each threshold (Nadir90)

| thresholds | recall | fpr | specificity | precision | F1-score | Youden's index |
|---|---|---|---|---|---|---|
| 0.00 | 1.00 | 0.95 | 0.05 | 0.04 | 0.08 | 0.05 |
| 0.01 | 0.96 | 0.26 | 0.74 | 0.12 | 0.22 | 0.69 |
| **0.05** | **0.90** | **0.10** | **0.90** | **0.26** | **0.40** | **0.80** |
| 0.10 | 0.85 | 0.06 | 0.94 | 0.34 | 0.48 | 0.78 |
| 0.20 | 0.76 | 0.04 | 0.96 | 0.45 | 0.56 | 0.72 |
| 0.30 | 0.70 | 0.02 | 0.98 | 0.53 | 0.60 | 0.67 |
| 0.40 | 0.64 | 0.02 | 0.98 | 0.60 | 0.62 | 0.62 |
| **0.48** | **0.60** | **0.01** | **0.99** | **0.66** | **0.63** | **0.59** |

| 0.50 | 0.58 | 0.01 | 0.99 | 0.68 | 0.62 | 0.57 |
| 0.60 | 0.51 | 0.01 | 0.99 | 0.74 | 0.61 | 0.51 |
| 0.70 | 0.46 | 0.00 | 1.00 | 0.82 | 0.59 | 0.46 |
| 0.80 | 0.39 | 0.00 | 1.00 | 0.87 | 0.54 | 0.39 |
| 0.90 | 0.31 | 0.00 | 1.00 | 0.94 | 0.47 | 0.31 |

Table S5 the performance analysis of the pre-dialysis IDH prediction model at each threshold (Fall20Nadir90)

| thresholds | recall | fpr | specificity | precision | F1-score | Youden's index |
| --- | --- | --- | --- | --- | --- | --- |
| 0 | 1 | 0.95 | 0.05 | 0.02 | 0.04 | 0.05 |
| 0.01 | 0.92 | 0.23 | 0.77 | 0.08 | 0.15 | 0.69 |
| **0.03** | **0.84** | **0.1** | **0.9** | **0.16** | **0.26** | **0.74** |
| 0.1 | 0.7 | 0.04 | 0.96 | 0.27 | 0.39 | 0.66 |
| 0.2 | 0.53 | 0.02 | 0.98 | 0.39 | 0.45 | 0.51 |
| 0.23 | 0.49 | 0.01 | 0.99 | 0.43 | 0.46 | 0.47 |
| **0.3** | **0.39** | **0.01** | **0.99** | **0.51** | **0.44** | **0.39** |
| 0.4 | 0.29 | 0 | 1 | 0.57 | 0.39 | 0.29 |
| 0.5 | 0.22 | 0 | 1 | 0.64 | 0.32 | 0.21 |
| 0.6 | 0.16 | 0 | 1 | 0.74 | 0.26 | 0.16 |
| 0.71 | 0.1 | 0 | 1 | 0.83 | 0.18 | 0.1 |
| 0.8 | 0.07 | 0 | 1 | 0.95 | 0.13 | 0.07 |
| 0.94 | 0 | 0 | 1 | 1 | 0 | 0 |

Table S6 the performance analysis of the intra-dialysis IDH prediction model at each threshold (Nadir90)

| thresholds | recall | fpr | specificity | precision | F1-score | Youden's index |
| --- | --- | --- | --- | --- | --- | --- |
| 0.95 | 0.00 | 0.00 | 1.00 | 1.00 | 0.00 | 0.00 |
| 0.80 | 0.05 | 0.00 | 1.00 | 0.83 | 0.09 | 0.05 |
| 0.70 | 0.10 | 0.00 | 1.00 | 0.79 | 0.18 | 0.10 |
| 0.60 | 0.17 | 0.00 | 1.00 | 0.67 | 0.27 | 0.17 |
| 0.50 | 0.24 | 0.00 | 1.00 | 0.59 | 0.34 | 0.24 |
| 0.40 | 0.31 | 0.00 | 1.00 | 0.52 | 0.39 | 0.31 |
| 0.30 | 0.40 | 0.00 | 1.00 | 0.44 | 0.42 | 0.40 |
| 0.27 | 0.43 | 0.01 | 0.99 | 0.42 | 0.42 | 0.42 |
| 0.20 | 0.51 | 0.01 | 0.99 | 0.33 | 0.40 | 0.50 |
| 0.10 | 0.66 | 0.02 | 0.98 | 0.23 | 0.34 | 0.64 |
| 0.02 | 0.89 | 0.07 | 0.93 | 0.10 | 0.18 | 0.81 |
| 0.01 | 0.93 | 0.11 | 0.89 | 0.07 | 0.13 | 0.82 |

| 0.00 | 0.95 | 0.14 | 0.86 | 0.06 | 0.11 | 0.81 |

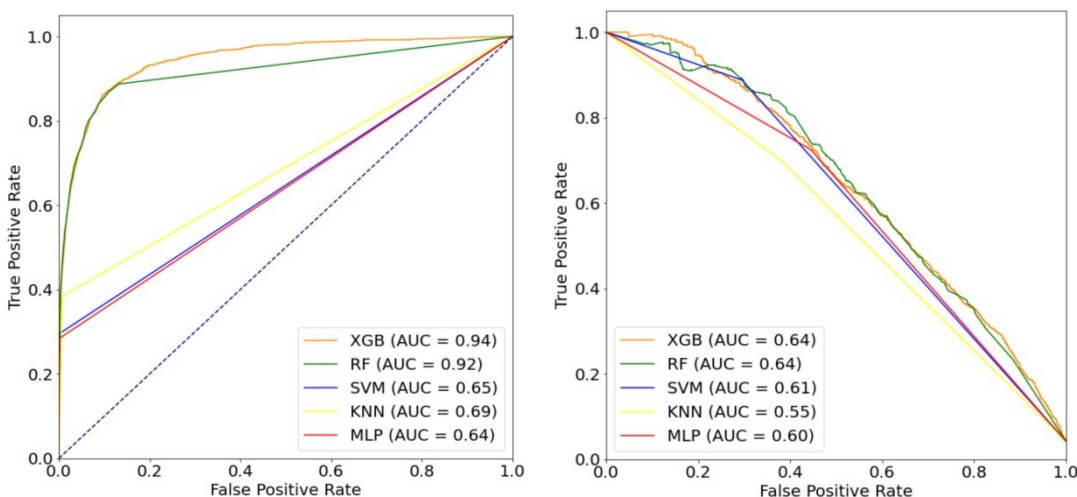

Figure S1 left:A Different models of ROC before dialysis; right:B PRC of different models before dialysis

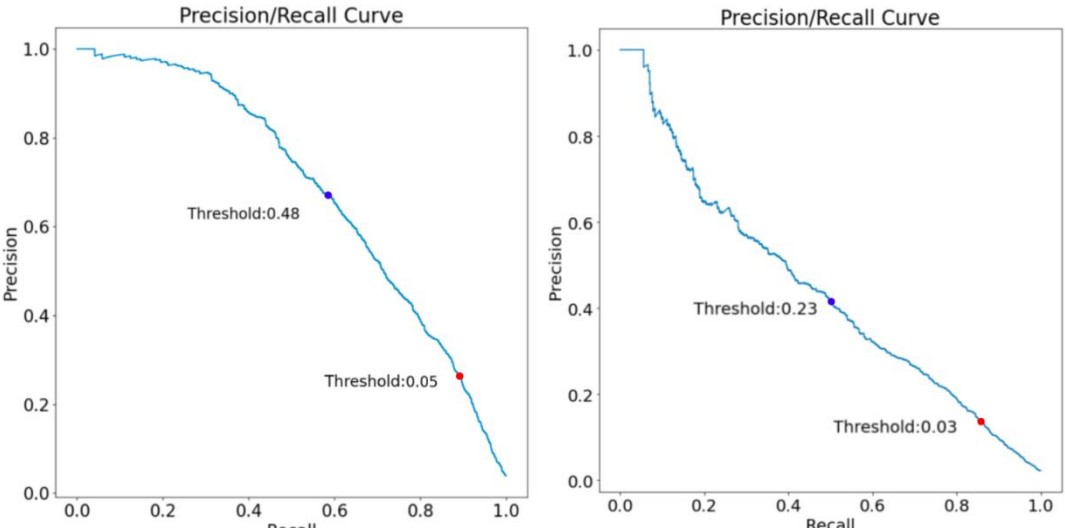

Figure S2  left:A PRC for Nadir90; right:B PRC for Fall20Nadir90

(The red and blue dots on the ROC and PRC curves respectively represent the threshold points corresponding to the maximum Youden index and maximum F1 value.)

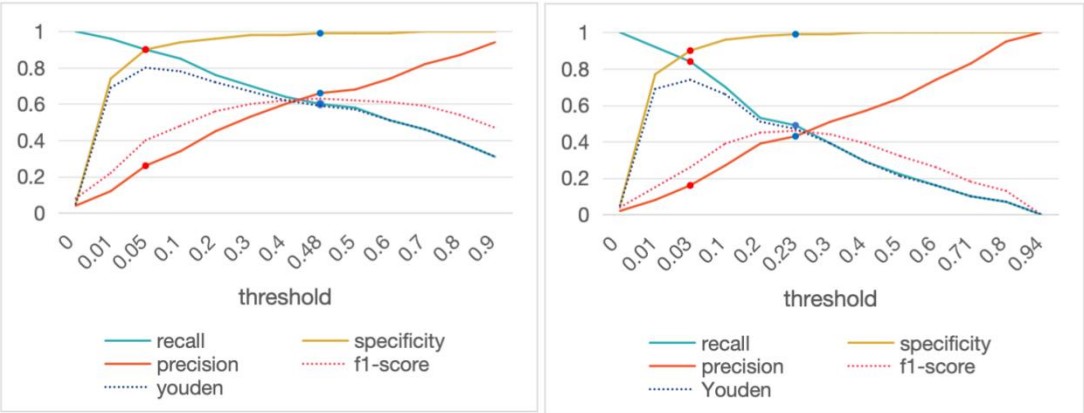

Figure S3 the performance analysis of the pre-dialysis IDH prediction model at each threshold (left:A Nadir90;right:B Fall20Nadir90)

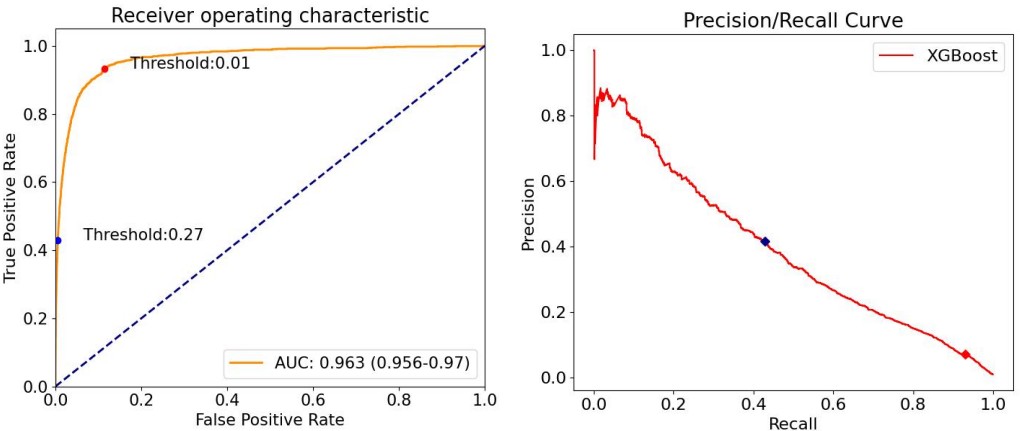

Figure S4  left:A ROC for Nadir90; right:B PRC for Nadir90

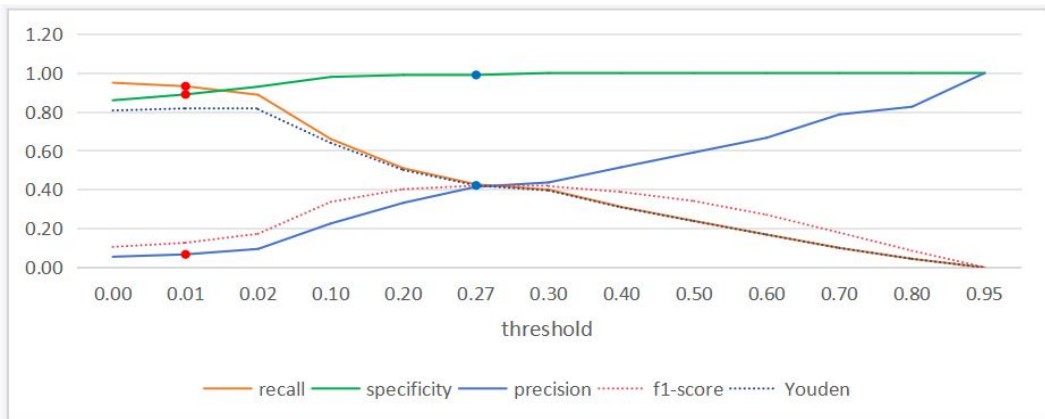

Figure S5 the performance analysis of the intra-dialysis prediction model at each threshold (Nadir90)

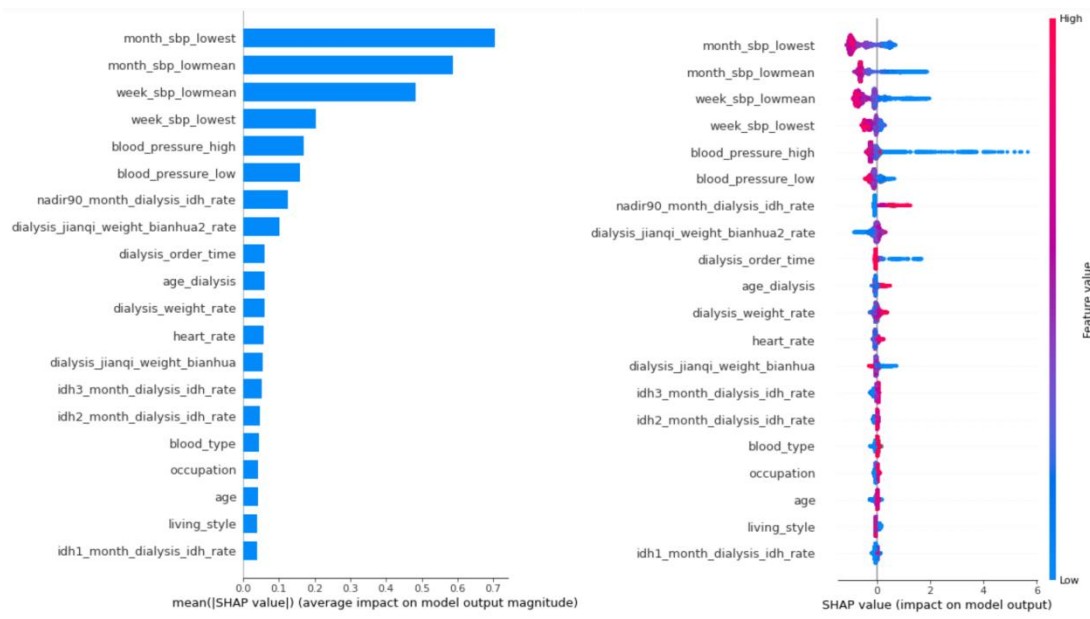

Figure S6 The XGBoost model predicts the top 20 important features of the pre-dialysis (Nadir90)

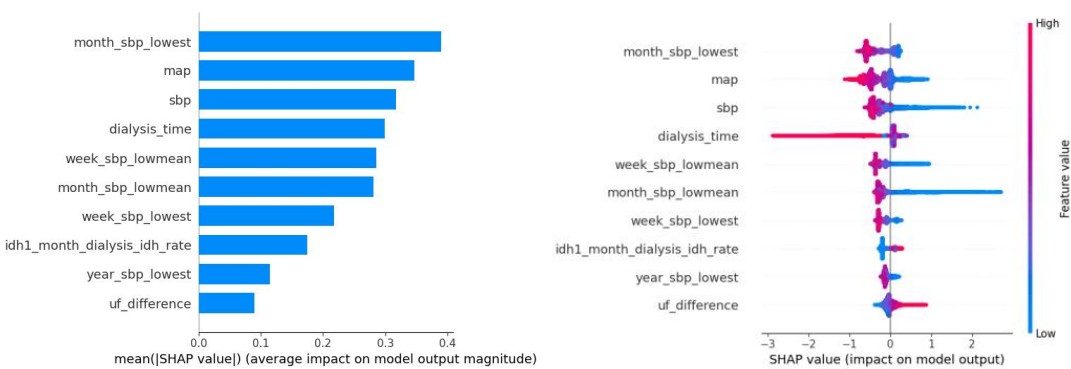

Figure S7  The XGBoost model predicts the top 10 important features of the intra-dialysis (Nadir90)