# OpenReview forum: "A Predictive Model for Real-time Prediction of Intradialytic Hypotension Based on Machine Learning Algorithms"
_KDD.org/2024/Workshop/AIDSH — KDD-AIDSH 2024 Poster_

### Official Review · Reviewer_s3U6 · 2024-06-12
**Good paper, accept**

**Rating:** 7
**Confidence:** 3

**Review:**

First of all, this submitted paper is not blindly.
Pros:
* Rigorous methodology: this study tilizes a range of machine learning models and cross-validation, with detailed evaluation with AUROC, AUPRC, TPR, etc.
* Clinical relevance: potential to improve patient outcomes by predicting IDH and enabling timely intervention.
* Novel approach: combining pre-dialysis and intra-dialysis data for real-time predictions is innovative.

Cons:
* Limited external validation: in this study, data were collected from a single center. The generalizability is uncertain.
* Incremental improvement: builds on existing work rather than introducing completely new concepts.
* Preliminary nature: needs more validation before clinical implementation.
Overall, this paper presents a robust and innovative approach to predicting intradialytic hypotension using machine learning algorithms, leveraging a comprehensive dataset and rigorous evaluation metrics. The authors have structured the paper clearly, providing detailed explanations and visual aids to enhance understanding. Despite the high complexity and dense technical content, which may pose challenges for non-experts, the novel combination of pre-dialysis and intra-dialysis data demonstrates significant clinical relevance. However, the study's limitations, including the lack of external validation and potential biases in data handling, indicate that further validation is necessary before clinical implementation. This work represents a substantial step forward in the intersection of healthcare and machine learning, offering promising insights while highlighting the need for broader applicability and additional research.

---

### Official Review · Reviewer_pmsC · 2024-06-18
**The study effectively develops a high-performing predictive model for intradialytic hypotension (IDH) using large-scale clinical data, achieving an AUROC of 0.955 with XGBoost. However, the innovation is limited by the application of existing methods without new theories, insufficient time-series modeling, lack of detailed ethical considerations, and inadequate handling of class imbalance.**

**Rating:** 7
**Confidence:** 4

**Review:**

## pros
- The study use of pre-dialysis and intra-dialysis data has significantly improved the accuracy of IDH prediction. The model not only considers pre-dialysis clinical variables but also utilizes real-time dynamic physiological data during the dialysis process, more fully reflecting the physiological changes in patients.
- The sample size is large, and the data are diverse. The study included 61823 dialysis records from 644 patients, with a total of 302942 systolic blood pressure measurements during dialysis, ensuring a substantial volume of data.

## cons
-  Although the manuscript has made some improvements in areas such as feature engineering, its overall innovation is not sufficient. The main contribution lies in the application of existing methods to specific problems, without proposing any new theories or methods.
-  The occurrence of IDH may be related to the patient's historical status and treatment process, exhibiting time-dependence. The paper does not adequately model and utilize time-series information. It primarily involves summarizing historical features through statistical aggregation, without considering the dynamics of time and sequential patterns.
- IDH is an event with a low incidence rate, and the data may have a serious class imbalance issue. The manuscript does not provide detailed descriptions of how the problem of sample imbalance was addressed, such as whether oversampling, under sampling, or cost-sensitive learning methods were used. Sample imbalance could affect the training and predictive performance of the model, necessitating a more comprehensive assessment.
- In the feature engineering section, the paper lists the raw features used and mentions the construction of some new features based on clinical knowledge. However, the paper does not provide detailed explanations of the specific definitions, calculation formulas, and clinical significance of each feature. For some complex derived features, there is a lack of clear construction logic. This could affect the reader's understanding of the features and interpretation of the results.
- The paper compares several machine learning models, but the description of hyperparameter tuning is somewhat brief. It does not specify the hyperparameter settings. Particularly, the parameters of the models used in the paper are quite important, and a more detailed discussion would be appropriate.

---

### Decision · Program_Chairs · 2024-06-28

Accept (Poster)